# Genetic Selection for Thermotolerance in Ruminants

**DOI:** 10.3390/ani9110948

**Published:** 2019-11-11

**Authors:** Richard Osei-Amponsah, Surinder S. Chauhan, Brian J. Leury, Long Cheng, Brendan Cullen, Iain J. Clarke, Frank R. Dunshea

**Affiliations:** 1Faculty of Veterinary and Agricultural Sciences, University of Melbourne, Melbourne, VIC 3010, Australia; rich12668@yahoo.co.uk (R.O.-A.); brianjl@unimelb.edu.au (B.J.L.); long.cheng@unimelb.edu.au (L.C.); bcullen@unimelb.edu.au (B.C.); iain.clarke@unimelb.edu.au (I.J.C.); fdunshea@unimelb.edu.au (F.R.D.); 2Department of Animal Science, University of Ghana, Legon, Accra, Ghana

**Keywords:** adaptation, climate change, food security, genetic markers, heat stress

## Abstract

**Simple Summary:**

Ruminants make important contributions to agricultural production, protein food security, livelihoods, and socio-cultural values, particularly in the developing world. Changing climate has dire consequences on animal agriculture and presents a real challenge for humankind. Increasing temperatures, solar radiation, humidity, and resultant heat waves, low rainfall, and drought compromise the availability of forage and water. These environmental factors adversely affect animal growth and reproduction and increase disease incidence as well as threaten biodiversity. The mitigation of such effects has been confined to location or breeds and is often expensive and not always sustainable in view of continuous variations in the climatic data. In this review we have proposed that genetic selection and breeding of thermotolerant ruminants provide a sustainable means of minimizing the effect of climate change on their production. Given the variation in the ability of ruminants to tolerate heat stress and the availability of genomic tools to pursue this agenda, heat stress can be minimised. This is a shared responsibility, requiring action by stakeholders across all sectors of society.

**Abstract:**

Variations in climatic variables (temperature, humidity and solar radiation) negatively impact livestock growth, reproduction, and production. Heat stress, for instance, is a source of huge financial loss to livestock production globally. There have been significant advances in physical modifications of animal environment and nutritional interventions as tools of heat stress mitigation. Unfortunately, these are short-term solutions and may be unsustainable, costly, and not applicable to all production systems. Accordingly, there is a need for innovative, practical, and sustainable approaches to overcome the challenges posed by global warming and climate change-induced heat stress. This review highlights attempts to genetically select and breed ruminants for thermotolerance and thereby sustain production in the face of changing climates. One effective way is to incorporate sustainable heat abatement strategies in ruminant production. Improved knowledge of the physiology of ruminant acclimation to harsh environments, the opportunities and tools available for selecting and breeding thermotolerant ruminants, and the matching of animals to appropriate environments should help to minimise the effect of heat stress on sustainable animal genetic resource growth, production, and reproduction to ensure protein food security.

## 1. Introduction

The Intergovernmental Panel on Climate Change (IPCC) has highlighted many negative effects of climate change arising from increased severity and frequency of drought, rainfall, floods, and high temperatures with huge consequences for the sustainability of global agriculture, producer incomes, producer livelihood, and food security [1]. The earth’s climatic data are expected to vary at very unusual rates with global warming expected to reach 1.5 °C by 2040 and average surface temperature predicted to increase by 1.88–4.08 °C (varies with different models and model assumptions) by 2100 [2]. A Commonwealth Scientific and Industrial Research Organisation (CSIRO) study [3] estimated that by 2025 average heat stress (HS) days with temperature humidity index (THI) > 75 will increase by 20–25 days, with consecutive HS days where THI > 75 increases by 2–4 in Victoria, Australia, relative to 1970–2000. Consequently, milk production losses of 35–210 litres/cow/year are expected in the absence of adaptation [3]. In Australia, during the years 2013–2017, there was a 27% upsurge in hot days (above 35 °C), and 2018 was Australia’s third-warmest year, with the summer of 2018–2019 being the hottest [4]. The Australian Bureau of Meteorology recently declared January 2019 as the hottest on record for the state of Victoria, and in the same month several states in the United States of America experienced their coolest-ever winter. Desertification, for instance, has reduced the carrying capacity of rangelands, forage quantity and quality, and increased livestock diseases, particularly in Africa, Australia, Central America, and Southern Asia with up to 50% loss of available biomass [5].Thus, climate change will continue to be detrimental to sustainable growth and livestock production, which is a major consequence in countries that depend on agriculture as the main source of livelihood for its citizens [6,7]. The World Economic Forum (WEF) Global Risk Report 2016 identified the lack of mitigation and adaptation strategies to climate change, natural disasters, and biodiversity loss among the top 10 risks faced by the World [8].

The need for more climate change adaptation research, especially in the developing world has been emphasised [9]. All efforts to increase livestock production to feed the growing human population [10] need to recognise this challenge and develop appropriate mitigation strategies [11]. This is crucial given the fact that majority of the population in the developing world depend on ‘rain-fed’ agriculture [12,13]. In addition, in well-integrated mixed farming systems, livestock provide draft power and manure, while crop residues are used to feed livestock [10]. Such mixed farming systems are commonplace in African agriculture, providing the staple diet of millions of people of low socio-economic status, with ruminant livestock production being the main source of income in most developing countries. For livestock to continue to play this role, ruminant livestock need to be bred that are more efficient by being able to adapt to low-quality feed and changing climate without compromising performance [14]. This review discusses the impacts of HS on sustainable animal agriculture, highlights the opportunities and challenges to genetically select for thermotolerance in ruminant livestock, and proposes sustainable breeding options for their production.

## 2. Climate Change and Animal Production

Climate change leads to rising temperatures, variations in photoperiod, and precipitation, leading to reduced or variable feed quality and quantity in most regions, less water availability, and higher disease susceptibility [15]. These factors lead to altered physiology and behaviour of animals, in an effort to adapt to harsh production environment and management conditions [16]. Body temperature is controlled by a balance between metabolic heat production and heat loss from the body [17] and when an animal is unable to adequately dissipate an excess of endogenous heat to maintain homeothermy [18], HS occurs. A thermotolerant animal is one that maintains thermal balance under conditions of heat load [19]. Heat stress compromises feed intake, growth, milk production and quality, and meat quality, resulting in a significant financial burden to global animal agriculture [20].

Intensive livestock production systems provide greater control over excursions in climate exposure so that HS may be minimised [21]. Unfortunately, such intensive systems are generally cost intensive and not a viable option for small-scale farmers in the developing world. In dairy cows, HS lowers milk production by between 25 to 40% due to reduced feed intake [22]. In dry cows, HS negatively affects the development of the mammary gland milk production in subsequent lactations [19,21,23]. Economic losses to the dairy industry are primarily associated with lowered milk production, reduced reproduction, increased metabolic disorders, and compromised immune function resulting from HS as a consequence of rising global temperature [15]. The economic losses in livestock production resulting from HS are substantial. For example, the United States alone was reported to lose USD 1.69 to 2.36 billion annually due to HS, of which USD 900 million was specific to the US dairy industry [24,25,26]. Though decreased milk production is the major contributor to the annual financial losses due to HS in the dairy industry, such losses can also be attributed to lowered reproductive efficiency, increased incidence of diseases, and higher mortality [27]. The immune response, which is compromised by HS, is a result of oxidative stress [28]. Thus, increased oxidative stress associated with reduced antioxidant status/increased production of free radicals (oxidants) has been observed in dairy cows [29,30], sheep [31,32,33], and poultry [34].

## 3. Interventions to Reduce Heat Stress on Ruminant Production

No single strategy can ensure the adaptation of ruminant livestock to climate change [35,36,37,38,39]. It has been reported, however, that no singular management measure or combination of measures has been able to totally eliminate the negative effects of a high heat load [40]. Although changes in the environment may relieve ruminants from thermal stress, the costs involved could lead to economic losses for farmers. In a review of mitigation and adaptation strategies, Zhang et al. [38] recommended different strategies for developed and developing countries as cost effective and locally available strategies have a better success rate of being adopted by local farmers [35]. For instance, whilst facility modification (provision of shades, sprinklers, and evaporative cooling services), changes in herd genetics, and choosing breeds with *Bos indicus* influence may be appropriate in developed countries, switching from the production of cattle to a smaller ruminant animal and increasing flock sizes will suit developing countries better. This is on account of the relatively heat-tolerant adaptive capacities of local sheep and goat breeds compared to cattle, their small sizes, better ability to forage on marginal lands, and their lower cost per unit of production. Additionally, sheep and goats adapt better to harsh environments and extreme diets and are tolerant to endemic diseases [41], making it easier for poor farmers in developing countries to keep them.

### 3.1. Environmental Modification and Nutritional Interventions

Genetic selection or development of thermotolerant breeds, modification of the environment, and nutritional intervention are key strategies to consider in the production of animal genetic resources (AnGR) in hot environments [42]. These strategies may provide an optimum productive environment for farm animals under various unfavourable climatic conditions. Additionally, changes in farming systems (cropping to grazing, adopting mixed crop-livestock systems, and reducing stocking densities) have also been used [38] by farmers. Farm management and infrastructure (cow cooling) have been modified to mitigate the effect of high THI on cattle production [43]. The ability to successfully cope with the effects of HS depends on farm location, livestock breeds, available appropriate technologies and agricultural advisory, or extension services [5]. Silanikove [44] recommends that under extensive conditions shade shelter should be provided, with water at vantage points to enable grazing animals to visit at least twice a day. Farmers should also avoid raising temperate breeds in hot conditions (ambient temperature > 25 °C; THI > 70) as well as transporting livestock from cold to hot environments in the summer [39]. Some HS mitigation strategies that have been reported are listed in Table 1.

### 3.2. Opportunities for Genetic Selection of Thermotolerant Ruminants

Most HS mitigation strategies hitherto have focused on housing to maintain optimum temperatures and humidity, variations in feeding, and reduction in stocking density. A key component of adaptation, however, is the genetic ability of an organism to survive under stressful conditions. Thus, genetic variation among animals (within and between breeds) is an important means of taking account of climate change. In the following sections, we explore opportunities and challenges of genetic selection as a tool for breeding climate-resilient ruminants for sustainable livestock production.

Environmental modifications and feeding strategies are production system-specific and their use mainly depends on their unique characteristics for improving the profitability of animal production. Genetic improvement, on the other hand, is more profitable as it produces a permanent and cumulative change [64] in the animal flocks/herds. Genetic selection for heat tolerance would provide a sustainable means of augmenting feeding and/or housing modifications [14,65]. In a review of such strategies, Gaughan [66] suggests to identify existing local breeds that are already adapted to production under environmental stresses and to allocate stress-adaptive genes in these breeds would provide a way forward. Subsequently, selection signatures for thermotolerance can be identified through functional genomics and productive breeds improved through cross-breeding with resilient genotypes and incorporation of stress-tolerant genes. For instance, the SLICK haplotype (http://omia.angis.org.au/ OMIA001372/9913/) originally identified in Senepol cattle has already been introduced into Holsteins to improve their thermotolerance. SLICK haplotype Holstein cows are better able to regulate body temperature, and experience less-pronounced reductions in milk yield under HS [67].

Metabolomics can also help detect key biomarkers for better heat tolerance (HT) as measurements of metabolites of HS in body fluids might not be too costly. The strategy here is to determine what metabolites are indicative of HT and how to use this information in the selection of heat-adaptive animals. Already a number of experiments have been conducted to measure some of these metabolites of HS in dairy cattle [68], beef cattle [69], dairy goats [70], and pigs [71]. For instance, significant correlations between metabolites such as lactate, pyruvate, creatine, acetone, β-hydroxybutyrate, trimethylamine, oleic acid, and linoleic acid in milk and plasma indicate that their concentration can give us an idea of HS-induced metabolomic alterations in blood [68].

#### 3.2.1. Diversity of Animal Genetic Resources (AnGR)

Tropical regions are characterized by relatively high HS environments compared with the temperate regions [72]. Local breeds, particularly those from the Near East and Africa are known for their better adaptation to high temperatures and harsh conditions [41]. In general, tropical and subtropical breeds have a greater adaptive capacity to stressful environments than exotics [73]. Zebu cattle (*Bos indicus*), for instance, have HT genes [74] and thus, cattle originating from zebu breeds are better able to regulate body temperature in response to HS compared to cattle from a variety of *B. taurus* breeds of European origin [44]. This resilience of Zebu is the result of their lower metabolic rate and increased capacity for heat loss [74]. *Bos indicus* breeds have been naturally selected for their low fasting metabolism (good survival ability) and consequently, comparatively low growth rate under good conditions [75]. This means that the *Bos indicus* breeds are better buffered against fluctuating feed supply than *Bos taurus* breeds which have an inherently higher appetite and gain/day under minimal stress compared to *Bos indicus* breeds [75]. In addition to THI, factors such as feed availability and quality, shelter, precipitation, disease challenges, market demands, and work force availability need to be taken into account [76,77] before introducing exotic germplasms to production systems. In this regard, matching local animal genetic resources (AnGR) to appropriate environments and their conservation will ensure genetic diversity and offer us more opportunities to genetically select and optimize breeds to changing climates or to replace populations hit by severe climatic events such as droughts and floods [78]. A high genetic diversity has been reported for most local livestock populations of developing countries [79,80,81,82,83], including cattle [80], using high-density single nucleotide polymorphism (SNP) data sets revealing selection signatures for adaptation to harsh conditions, hot conditions, and pathogen pressure [74].

Temporal *HSP70* gene expression is a biomarker for adaptative gene discovery in cattle and has significant implications in the development of heat- and cold-tolerant genotypes in the context of climate change. Variations in the expression pattern of the HSP70 family and other HSP genes in different seasons may be central in potential strategies to breed for better adaptability in cattle [37]. Genetic polymorphisms in the sodium/potassium-transporting ATPase subunit alpha-1 *ATP1A1* enzyme gene and their association with HT have been reported in crossbred Jersey [6]. Heritability and genetic correlations from this study provide evidence of the possible selection for both milk production and HT. One genotype at locus C1787061T has been found to improve heat tolerance and total milk production in Sahiwal cows indicating association at this *HSP90ab1* SNP [84].

Collier et al. [85] reported on the groups of genes involved in bovine HS responses. Among these were changes in the expression of genes involved in increased glucose and amino acid oxidation as well as reduced fatty acid metabolism, endocrine system activation of the stress response, and immune system activation, which ensure extracellular secretion of HSP. Genomic analyses have also revealed the existence of gene variants related to hair and skin properties, immune responses, the nervous system, and tick resistance [86,87]. The complexity of the response to HS is associated with a number of mechanisms involving different genes [85], including those associated with hair and coat characteristics, cellular response, and those acting at the systemic level [85,86,87,88,89,90]. Olson et al. [90] found a major gene (SLICK) associated with HT with a dominant inheritance determining a type of hair, namely, ‘slick hair’. The gene has been introduced into Holstein–Friesian cattle, conferring thermotolerance [17,67,88]. This dominant gene on BTA20 is associated with RT [89,90]. According to a 2018 report on research by the University of Florida (https://www.dairyherd.com/article/new-heat-tolerant-holstein-genetics-available), the SLICK haplotype is a dominant trait that produces cattle with a short, sleek hair coat due to a mutation in the prolactin receptor gene. Holstein cattle possessing the SLICK haplotype have been created by the University of Florida. Two mutations of SLICK have been reported, one common to Senepol and Romosinuano cattle, and the other in Corora cattle [89].

In a review, Rolf [45] lists pathways and genes that have been identified in genomic studies as potential candidate genes for HT in cattle. A particular study of African cattle reported positively selected candidate genes for HT, including genes for oxidative stress, osmotic stress, heat shock, sweating, hair coat type, coat colour, feed intake, energy, homeostasis, and reproduction [74]. An extensive review, [81] has listed genes involved in heat tolerance in small ruminants. Significant associations have been found between the *ATP1A1* and sodium/potassium-transporting ATPase subunit alpha-2 *ATP1B2* enzyme gene polymorphisms, HT, and respiratory rates in dairy cattle [6,66,91]. Heat shock protein gene (*HSP70* and *HSP90*) response is regarded as a cellular thermometer for HS and other environmental stressors. Heat stress-tolerant (HST) dairy cows exhibited higher *HSP70* mRNA level expression than HS-susceptible (HSS) individuals in different breeds [7,37,83,84,92,93,94]. Additionally, the expression of *HSP70* gene in cold-adapted goats during summer and in heat-adapted goats during winter has been relatively higher [7,37,83,84,92,93,94]. Genes involved in signalling pathways that are directly or indirectly associated with thermotolerance (e.g., *MTOR and MAPK3*) have been found in ruminants. Several candidate genes were identified, which are associated with adaptation to thermal stress; the homeobox genes, *HOXC12* and *HOXC13*, play a role in hair follicle differentiation, growth, and development by regulating keratin differentiation-specific genes [74,95]. Selecting heat-tolerant animals under a high production efficiency will be an effective management method for HS [96,97]. However, it is recommended that one selects for HT within a high milk-producing breed rather than select for high milk production in a breed that is highly adapted to hot climates, due to the increased number of generations required for the adapted breed to reach optimum production levels [45]. The quantification of HT is fundamental if it is to be considered a potential selection goal in breeding programs [98]. Quesada et al. [99] indicated the importance of the tolerance and adaptive capacity of various breeds as a technical basis for selection of sheep and their use in various crossbreeding programs. The use of breeds that evolved in hot climates, or gene introgression from hot-climate breeds into populations of temperate breeds, has met with some success [40,67]. The selection for HT in high milk-producing dairy breeds and their incorporation in breeding programmes could be equally effective by avoiding the negative association between HT and production traits and may deliver faster results given the recent advances in genomics. Various studies have found that HT is positively correlated with fertility and has a strong negative correlation with production, but the underlying genetic interactions are not well understood [82,97,100].

#### 3.2.2. Availability of the State-of-the-Art Omics Technology

The availability of modern state-of-the-art tools and technologies to generate more thermotolerant phenotypes and to identify thermosucceptible animals may allow simultaneous selection of thermotolerant productive animals [40]. Once such genes in zebu cattle or *Bos indicus*, in general, are isolated, appropriate breeding strategies can be applied to further exploit them for climate smart production [101]. Modern phenomics, genomics, and transcriptomics technologies now abound to enable accurate selection of heat-tolerant animals without compromising milk production [40]. Advances in the use of sequence data as well as results of gene expression studies can lead to persistence of genomic breeding values across breeds and generations. Information from gene expression or genome-wide association studies (GWAS) can be used to further improve the accuracy of selection. Genome-wide association studies have often been used to identify regions of the genome that have a specific effect on a trait of economic importance in ruminant livestock [102].

Genomic selection presents the advantage of accelerated genetic gain, as livestock can be selected from a young age using their estimated genomic breeding values (GEBVs), rather than waiting to test their progeny [30,103]. Heat tolerance genomic breeding values can be predicted with an accuracy of between 0.42 and 0.61 using high-density SNP genotypes. In 2017, for the first time, breeding values (BVs) for heat tolerance were released in Australia [103]. Genomic estimated breeding values (GEBVs) for HT for Australian Holsteins and Jerseys [104] are of relevance because profitability and animal welfare can both be improved by identifying animals that are able to adapt to current and future climate challenges [105]. Genomically predicted heat-susceptible and heat-tolerant animals show significant differences in milk yield loss and rectal and intravaginal temperatures when exposed to a mild simulated heat wave [30]. Additionally, genetic gain is permanent, persisting through subsequent generations [40,104]. Moreover, the addition of GEBVs for HT should become relatively inexpensive, as increasing numbers of cattle are genotyped [104]. However, because HT is a genomic trait that has been identified only recently, the GEBVs are less reliable than those of already established phenotype-based production traits [103]. The accuracy of GEBVs for HT is further limited using variations in test-day milk production to measure heat stress, as variations between days are not wholly dependent on ambient conditions [104,105,106]. Data on core body temperature with increasing temperature and humidity as well as altered fatty acid profiles in milk using mid-infrared spectroscopy (MIR) may improve the accuracy of GEBVs [103].

Reducing the severity of HS in dairy cattle by genetic selection for increased thermotolerance is a largely unexplored strategy. The rectal temperature of animals under HS is heritable [82], so genetic selection for superior individuals based on this trait could reduce the effects of HS on production through increased thermotolerance [107]. A GWAS of rectal temperature (RT) under HS conditions in lactating Holstein cows to identify SNPs associated with genes for thermotolerance has been undertaken and this should aid in genetic selection for adaptation to HS [107]. There is considerable genetic variation for HT in Holsteins [82] and specific SNPs associated with genes for HT have been identified [96,106].

Signatures of selection in small ruminants (sheep and goats), indigenous to hot arid environments, have been studied using genome-wide SNP scans and several candidate regions underlying adaptation identified [108]. This provides a basis for investigating ruminant evolution and functional genomics in different species surviving in a similar environment [108]. Some candidate genes, that are linked to the adaptation of small ruminants, such as genes coding for growth hormone (*GH*), growth hormone receptor (*GHR*), insulin-like growth factor (*IGF-1*), leptin receptor (*LEPR*), and thyroid hormone receptor (*THR*) have been reported [109]. Heat shock factor 1 (HSF1), heat shock protein gene 60 (*HSP60*), *HSP70*, *HSP90*, and ubiquitin are linked to the resilient capacity of small ruminants, particularly under HS challenges. Among these thermotolerant genes, *HSP70* is a well-known genetic marker for thermo tolerance in small ruminants. Identification of such cellular and molecular markers may contribute to efforts to develop climate resilient breeds [110]. Phenotypic groups of Santa Ines sheep differ in heat tolerance with white sheep displaying more heat-resistant traits. Animals with longer hair and thicker and darker coats are more stress-susceptible, with wool sheep also showing less adaption to tropical climates [110]. In sheep, breed variation has been reported regarding response to HS [94,111] with breeds with less thermal insulation using less physiological resources to dissipate heat.

In general, hair sheep breeds adapt to HS better because of low concentrations of thyroid hormones and their relatively low metabolic heat production and slower breathing compared to wool breeds [36]. McManus et al. [110] reported that these breeds have skin and coat characteristics which facilitate their adaptation to HS. In general, Dorper × Pelibuey ewes have been found to have a high adaptive capacity to summer HS because they maintain rectal temperatures in the normal range and have a greater efficiency in losing body heat during the day by increased respiratory rate and skin temperature [112]. Furthermore, reduced kidney function in summer may indicate activation of mechanisms to prevent dehydration, which assists in the maintenance of homeothermy [112].

Three major responses seem to be elicited at the molecular and cellular level in response to HS. Initially, there is high expression of molecular chaperones and heat shock genes that prevents protein aggregation and misfolding, thus promoting cell survival. Then, the extracellular presence of HSPs activates the immune system. Eventually, continued exposure to severe HS leads to expression of cell cycle arrestors and tumour suppressors as well as genes involved in apoptotic signalling. The association of the genes involved in this process could be exploited to select and breed animals with improved HT [113]. In cows, markers linked to sensitivity of milk production to feeding level and sensitivity of milk production to THI have been identified on BTA9 and BTA29, respectively, and validated in two independent populations originating from different cattle breeds [107]. These validated marker panels should help in genetic selection for high milk production under HS conditions. Signatures of selection including genes controlling anaemia and feeding behaviour in the trypanotolerant N’Dama cattle, coat colour and horn development in Ankole cattle, and heat tolerance and tick resistance found across African zebu cattle breeds [80] provide other opportunities for genetic selection.

### 3.3. Challenges of Genetic Selection

As discussed in the previous sections, genetic selection of resilient and productive ruminant livestock could provide a major fillip or boost to livestock production in the 21st century. Genetic selection of animals adapted to HS should help increase production throughout the hot season and an important initial step in this process is to identify genes that specifically respond to HS. While significant progress has been made in this area, there is need for further research on genetic selection of the desirable animals [78]. Limitations also abound particularly in developing countries to enable the implementation of genomic selection and involve human, institutional, logistical, and financial challenges [76,114,115]. Furthermore, there is a need to investigate the genetic mechanisms underlying both cold and heat stress to ensure that resilient animals are bred for both conditions.

AnGR in developing countries have over the millennia survived and adapted to most of the negative effects of climate change because of their innate adaptation, but the greater demand for human food and the imperative to increase producer incomes have led to indiscriminate crossbreeding, which has diluted the adaptive germplasms [116]. This latter point is important because, if adaptive genotypes disappear, the populations required for selection programs will not be present. There is an urgent need for all countries to implement the global plan of action for AnGR [117] and to put in place appropriate conservation schemes for all local AnGR and sustainable livestock breeding programmes.

Another challenge in our bid to select for resilient ruminant livestock is the complexity of the mechanism underlying heat stress response and its genetic antagonism with production [40]. In consideration of the effects of HS on dairy cows, West [118] concluded that selection of heat-tolerant animals is possible and recommended that an appropriate and sustainable strategy be employed to prevent the increasing susceptibility to high heat loads associated with high milk-producing dairy breeds. Nevertheless, the selection of cattle that are genetically resistant to HS continues to be a challenge because of the known negative genetic correlation between heat tolerance and milk yield [82]. However, a lot of genetic variability exists in an individual animals’ response to increased HS, with a moderate genotype x environment interaction, which means that the best producers under thermal comfort may not be the best under HS [40]. Bernabucci et al. [18] investigated the effects of HS in Holstein cattle and found that there was an additive genetic effect over parities and their data confirmed the negative association between milk production and HT. Consequently, the identification of genetic markers which are not negatively associated with milk yield significantly will be an appropriate selection criterion [119]. Additionally, breeding strategies to improve HT will also depend on the production system with those able to provide enough resources for high productivity gaining more from such interventions. On the other hand, production systems with scarce resources will benefit from crossing with local (resilient) stock [19]. Most parts of the developing world lack both the institutional and human resources to overcome this challenge unless changes in policy and adequate financial support for research and training are provided.

In sheep, as in dairy cattle, milk yield is also negatively associated with heat tolerance, and selection for high milk production alone may reduce HT [120]. The use of heat-resistant individuals in sheep breeding programs thus has been proposed to improve animal welfare and productivity in hot climates. As in dairy cows, genetic variability and a moderate genotype x environment interaction have been observed in sheep [120,121,122,123].

A further challenge in the generation of heat-tolerant animals is to identify the right means of quantifying HS–HT phenotyping. According to [124], the use of available production and meteorological data to produce measures of HT captures only a small fraction of variability due to heat stress, because recording variations are not considered. It is important to properly and accurately collect phenotypic information which will reflect the level of stress. THI considers the combined measurements of temperature and humidity in which animals can best survive [20]. Body temperature is frequently recognized as the gold standard measure of heat tolerance, but its recording has been impractical for large applications. Nevertheless, advances in precision farming with respect to measurements of body temperature may open new possibilities to introduce body temperature in selection programmes [124]. If the THI threshold is reached and exceeded, it can lead to both physiological and behavioural changes in cows, as they attempt to maintain a constant body temperature, compromising production outputs such as fertility and milk production [30,121]. Although the THI is often used to gauge the extent of HS because of its ease of computing, it is an environmental parameter and does not consider thermal radiation, wind speed, or the duration of exposure to these conditions [122]. Thus, it has been suggested that THI does not adequately describe the effect of climatic conditions on livestock. Furthermore, except for the case of cattle, THI has not been specifically developed for other livestock and some specific breeds in certain geographical areas. For example, in goats, the few studies carried out so far have used THI formulations developed for beef and dairy cattle, with very few exceptions [125]. Apart from THI, the heat load index (HLI) [122] and the dairy heat load index (DHLI) [19] models are proposed, but these require careful validation based on stock characteristics, management practices, and mitigation variables [20].

Genomic selection for HS would be a sustainable tool to help reduce management costs and low production, usually associated with high environmental temperatures. Moreover, current breeding values of new traits that rely on a dedicated reference population have lower reliabilities and when these are added to a national selection index, there is erosion of overall reliability. Advances in genomic prediction methodology and regular validation should alleviate some of these issues by being instrumental in increasing reliabilities [103]. Schefers and Weigel [123] conclude that accuracy is a function of the reference population size which was used to estimate SNP effects. Thus, depending on how many cattle are available for screening, the accuracy may vary between herds, breeds, and countries. An option to increasing the reference population would be to include a wide variety of genotypes as much as possible from different environments. Even though genomic selection allows a reduction in the generation interval, it is possible that inbreeding rates per year could increase. Therefore, one needs to sustainably use and conserve genetic diversity of current and potential progeny via genome-based mating programmes [123].

Finally, the cost of genomic selection is an important issue; if this is too high, farmers will be disinclined to participate in a breeding scheme. In 2011, the cost of genotyping was approximately AUD 100 per heifer and was proven to be not economically viable as a selection tool [126]. If the cost of genotyping was reduced (perhaps by subsidy), this would have a major impact on animal profitability and production. It is encouraging that the cost of genotyping of cattle has shown a steady reduction [127], which will make genome-wide selection more affordable in the future.

## 4. Implications and Recommendations

There is an urgent need to support future research aimed at ameliorating HS in livestock including:-the development and monitoring of appropriate phenotypic indices of HS;-the assessment of genetic factors associated with HS by use of genomics and proteomics;-the development of innovative and more sustainable management practices to reduce HS and improve animal welfare and production;-the assessment of epigenetic consequences of climate change.

Epigenetic effects may explain differences in adaptive potentials that have evolved over generations and provide new information to elucidate the molecular and cellular mechanisms of ruminant adaptation [128]. Overall, it is essential that the opportunities of animal genetic diversity be secured. This requires:-better identification and characterization of breeds and their production environments;-inventory, monitoring, and establishment of early warning systems for breeds to help monitor and respond to threats to genetic diversity;-establishment of more effective and sustainable in situ and ex situ conservation measures;-genetic improvement programmes owned by key stakeholders and targeting adaptive traits in high-output and performance traits in locally adapted breeds;-human and institutional capacity building for developing countries in their management of animal genetic resources;-wider access to genetic resources and associated knowledge [39].

### 4.1. Matching Appropriate Genotypes to Production Environments and Systems

Climate change has come to stay with us and therefore it is important to understand how farmers perceive and adapt to climate change. Such data will help in the development of future adaptation and mitigation strategies [129]. Future research must also examine the possibility of adopting cattle breeds that are better equipped for hot environments, thus minimizing the duration of compromised welfare under HS [25]. The choice of the breeds and breed composition in crossbreds should be adapted to the local agroclimatic environment and socio-cultural context, giving priority to animals that cope well with harsh climatic, nutritional, and health conditions. The few examples of genomic selection tend to favour crossbreeding and one of the best practices to ensure the sustainability of such a strategy is to maintain purebreds at all times [130], as for example reported from Bangladesh [131]. The availability and conservation of all available genetic diversity and breeding alternatives allow the preservation and improvement of purebred indigenous populations, thus exploiting their specific adaptive features, together with the local production and dissemination of crossbreds [130].

### 4.2. Redefining Breeding Policy and Objectives

Current climate change adaptations and their tailor-made mitigation practices as well as policy frameworks are critical to sustain livestock production [132]. Appropriate adaptation strategies are needed to improve the resilience of crop and livestock productivity to climate change [133]. Whilst mitigation measures can reduce the negative effects of climate change on livestock, these measures can only make a significant impact if adopted into national and regional policies [10]. There is a need to also improve our knowledge of especially low-input production systems in order to gauge how they may change and adapt as a result of climate change, and how government policy can most effectively provide an enabling environment required for sustainable production systems [134]. Livestock producers as key stakeholders should have a say in the choice of appropriate and sustainable adaptation and mitigation strategies [66]. Animal breeding organizations must collaborate with other disciplines and stakeholder institutional partners such as agronomists, physicists, meteorologists, engineers, and economists. The effort in selecting animals that, until now, has been primarily oriented toward productive traits, must be oriented toward robustness, in the future and above all must include adaptability to HS. In this way, selective breeding would eventually increase the frequency of thermotolerant genotypes with much needed phenotypic characteristics. Research must continue to develop new and more innovative, cost effective, and appropriate techniques, particularly those requiring low energy expenditure. Species- and breed-specific phenotyping tools and indices that are more robust than THI must also be developed, to guide farmers and other stakeholders [5].

### 4.3. Institutional and Human Capacity Building

Reducing the adverse impacts of climate change on livestock and breeding for thermotolerance requires multidisciplinary approaches, including the integration of animal breeding, physiology, nutrition, housing, and health [66], among others. Therefore, we need to invest in human and institutional capacities and also more North–South cooperation to share valuable information and reduce costs. Successful analyses and interpretation of genomic data require appropriate computer hardware and software as well as expertise in database development and support, quantitative genetics, and statistical modelling. The establishment of genetic improvement programs is worthwhile only when the production environment is well resourced in terms of adequate feed, shelter, and veterinary care for animals. Consequently, there is a need for stakeholders, including the farmers, to be trained in animal nutrition, health, and management to improve animal welfare as well as record keeping and analyses. Finally, in terms of human capacity development, training programs for scientists and managers of breeding programs are needed in quantitative genetics, genomics, and bioinformatics as well as skills in accessing and processing online genomic databases and scientific literature resources [76,115,135]. South–South and North–South co-operations are to be encouraged to facilitate such training [130]. Indeed, this review is a product of one such collaboration.

## 5. Conclusions

Improved knowledge of the physiology of ruminant acclimation to harsh environments, the opportunities and tools available for selecting and breeding thermotolerant ruminants, and the matching of animals to appropriate environments should help to minimise the effect of heat stress on sustainable animal genetic resource growth, production, and reproduction to ensure protein food security. This review clearly demonstrates that genetic selection and breeding of thermotolerant ruminants is a viable option to minimize the effect of climate change on their production. This is justified because of the established variation in the ability of ruminants to tolerate heat stress and the availability of human and institutional capacity the state-of-the-art genomic tools to facilitate this objective. However, the investments and commitment needed to achieve this goal is a shared responsibility, requiring action by stakeholders across all sectors of society. 

## Figures and Tables

**Table 1 animals-09-00948-t001:** Mitigating strategies against Heat Stress in ruminants.

Mitigating Strategy and Reported Effect	Species	References
Provision of shade/shelter—shade reduced high heat load by 30% or more; shaded animals had relatively lower rectal temperatures and rates of respiration as well as increased milk production; shade also reduced the effects of HS on faecal cortisol metabolites.	Dairy cattle	[20,45,46,47,48,49]
Provision of sprinklers, fans, and misters—these provide evaporative cooling and convection; cows under sprinklers had a marked lower body temperature than those in shade alone and remained lower for at least 4 h after milking.	Dairy cattle	[49]
Perforated air ducting (PAD) systems—The PAD system lowered air temperature by 1.5 °C and increased relative humidity by 8.1%. The rectal temperatures of cows under PADs were significantly lowered after 15 days and they produced more milk.	Dairy cattle	[50]
Increasing concentrate and decreasing forage content of the diet—this increased the energy density of the diet, whilst reducing intake and heat of digestion; feeding maize instead of wheat-based diets also provided a similar effect.	Dairy cattle	[20]
Feeding corn grain plus forage or sodium-treated wheat forage reduced heat production—wethers that were fed corn grain-based forage (CD) had relatively lower respiratory rates (RR) and rectal and skin temperature; wethers that were fed 3% NaOH-treated wheat plus forage (TWD) had lower RR and flank skin temperature (FT) than wethers that were fed wheat grain plus forage (WD) during HS.	Sheep	[51]
Feeding corn grain plus forage led to a low rate of fermentation and the least heat production—wethers that were fed CD had lower RR, rectal temperature (RT), left flank skin temperature (LFT), and right flank skin temperature (RFT) than WD-fed wethers, and this benefit was the greatest during HS. Feeding CD may help reduce HS in sheep.	Sheep	[52]
High dietary Vitamin E and selenium (Se) supplementation—sheep on high Se and high Vit E diet had lower rates of respiration (191 vs. 232 breaths/min; *p* = 0.012) and rectal temperature (40.33 °C vs. 40.58 °C; *p* = 0.039) under peak HS (1700 h). Supranutritional dietary Se or Vit E can reduce some of the negative effects of HS.	Sheep	[31]
Vit E supplementation—Vitamin E supplementation in the diet of periparturient dairy cows led to increased plasma and milk vitamin E and decreased somatic cell count in milk.	Dairy cattle	[53]
Antioxidant supplementation (Vit C and Vit E with Se)—rectal temperature and respiratory rates increased (*p* < 0.05) in HS group with improvement in acid-base status of the antioxidant-supplemented groups.	Goats	[54]
Betaine supplementation—betaine supplemented at 2 g/day ameliorated HS responses and may thus have beneficial effects for sheep exposed to heat.	Sheep	[55,56]
Chromium (Cr)—chromium supplementation improved heat tolerance (HT) in heat-stressed animals	Dairy cattle	[23,57,58,59]
Niacin (Vit B_3_)—niacin increased HT in cattle by increasing evaporative heat loss in vivo and cellular heat shock response by increasing the gene expression of heat shock protein (HSP) 27 and HSP70 during thermal stress in vitro.	Dairy cattle	[60]
Lipoic acid—lipoic acid enhanced insulin action in animals and therefore may improve HT and animal performance.	Chickens, swine, horses, ruminants	[23,61,62]
Thiazolidinediones (TDZs)—TDZs could be useful for improving and ensuring glucose use and upregulating HSPs in heat stress conditions on account of improved insulin action.	Dairy cows	[23]
Vit C, Vit E, trace mineral (zinc and selenium), and electrolyte supplementation—all these have positive effects on heat-stressed animals.	Ruminants	[63]

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
