# Peer review of "Genetic Selection for Thermotolerance in Ruminants"

_animals, 2019, doi:10.3390/ani9110948_

Round 1

Reviewer 1 Report

The authors have complied with most of the suggestions made. I consider that the manuscript is suitable for publication in its present form

Reviewer 2 Report

Osei-Amponsah et al. provide a comprehensive review on the genetic selection for thermotolerance in ruminants. The article is well written and easy to follow. I only have minor comments, which should be addressed before publication.

L35: should help to minimise

L47: I'm not familiar with the abbreviation CSIRO, so maybe the authors can provide more information

L51: Use % instead of per cent

L82: At the beginning of a sentence do not use abbreviations

L87: cost intensive

L102: a high heat load

L112: remove both

L146: suggest to identify

L147: to allocate genes

L159: replace cow milk with dairy cattle

L167: Thus, cattle originating form zebu breeds

L168: replace than are with compared to

L170: replace automatically with naturally

L174: compared to

L178: to genetically select and optimize breeds

L181: using high-density

L187: may be a central

L196: activation, which

L205: use abbreviation for bovine chromosome (BTA)

L206: RT and RR?

L210: have been selected/created?

L226: are directly or indirectly associated

L233: Please check the application of abbreviations heat tolerance (HT?), throughout the manuscript

L249: heat-tolerant (HT?)

L257: Do not start a sentence with an abbreviation, have instead of has

L261: see previous comment

L262: remove of, In 2017 for the first time breeding.....

L263: please make sure the meaning of genetic selection vs. genomic selection

L298: Identification of....

L302: ,whilst wool sheep 

L322: involved in this process, remove for

L324: on BTA 9 and BTA 29

L325: originating from different cattle breeds

L335: in this process

L389: for other livestock

L392: other models are proposed

L439: remove of

Reviewer 3 Report

    1."Thus, climate change will continue to be detrimental to growth growth  and livestock production, which is a major consequence in countries that depend on agriculture as the main source of livelihood". It is recommended to supplement the impact of climate change on other species (such as poultry, pigs, and sheep) before this sentence.

In part of “2. Impact of climate change on animal production”, it is also recommended to supplement the impact of climate change on other species. “A key component of adaptation, however, is the genetic ability of an organism to survive under stressful conditions. Thus, genetic variation among animals (within and between breeds) is an important means of taking account of climate change. In the following sections, we explore opportunities and challenges of genetic selection as a tool for breeding climate resilient ruminants for sustainable livestock production.” It is recommended to move this text to 3.2. Line 159-160: Please add specific metabolites.

Author Response

This manuscript is a resubmission of an earlier submission. The following is a list of the peer review reports and author responses from that submission.

Round 1

Reviewer 1 Report

The authors have done a comprehensive and discerning review of the possibilities for genetic selection of heat tolerance in livestock. Minor suggestions follow.

General comments

An unmentioned challenge is that cold and heat stress are coexisting problems in some regions of the planet and genetic mechanisms for both types of thermal stress seem to differ. For example, reducing thermal insulation may not be adequate in those regions.

Another challenge is heat tolerance phenotyping. The use of the available productive recording and meteorological data to produce measures of heat tolerance captures only a small fraction of variability due to heat stress because no consideration is given to variability between recordings (Misztaletal., 2006. Issues in genetic evaluation of dairy cattle for heat tolerance. 8th World Congress on Genetics Applied to Livestock Production, August 13-18, 2006, Belo Horizonte, MG, Brasil). Body temperature is frequently recognised as the gold standard measure of heat tolerance but its recording has been impractical for large applications. Nevertheless, advances in precision farming with respect to measurements of body temperature may open new possibilities to introduce body temperature in selection programmes

The use of metabolomics as a way to point at key biomarkers of heat stress or better of heat tolerance might also want to be mentioned. Measurement of metabolites of HS in body fluids (using for example milk MIR spectra) may be not too costly. There are already a number of studies of metabolmics of HS in livestock that may want to be mentioned. Some examples are Tian et al., 2016 (doi.org/10.1038/srep24208 ) for cow’s milk, Qu and Ajuwon, 2018 (doi: 10.1093/jas/sky127) in pigs, Liao et al., 2018 (doi.org/10.1021/acs.jafc.8b01794 ) in beef cattle, Contreras et al. ,2019 or (doi.org/10.1371/journal.pone.0202457), in dairy goats. The challenge here is to determine what metabolites are indicative of heat tolerance and how to use this information in the selection of heat tolerant animals.

Specific comments

L 30-1. “One effective strategy may be sustainable intensification of ruminant production. “ What do you mean by sustainable intensification? One might guess that you mean heat abatement strategies, but you may want to be clearer here.

Ll 47-8. “the number of consecutive HS days …” the number of consecutive days where THI>75? Please be more specific.

L 89. Please, change “HS. [20]” by HS [20].

Ll 432-43. “Genomic selection for HS would mean that farmers would not have to endure the increased management costs and reduced production”. This seems to me a too optimistic vision of genomic selection for HS. From my point of view, genetic selection (genomically enhanced or not) is one more tool to deal with heat tolerance but I am not sure that it will totally eliminate declines in production or the need for some type of heat abatement.

Reviewer 2 Report

Animals 2019

Manuscript ID: animals-566406
Type of manuscript: Review
Title: Genetic selection for thermotolerance in ruminants
Authors: Richard Osei-Amponsah, Surinder Chauhan *, Brian Leury, Long Cheng,
Brendan Cullen, Iain Clarke, Frank Dunshea
Submitted to section: Animal Genetics and Genomics, Systems.

General comments

There are conceptual deficiencies, e.g. the MS proposes (e.g. lines 16-18) “that genetic selection and breeding of thermotolerant ruminants provides a sustainable means of overcoming climatic influences on production.” Thermotolerance, i.e. the capacity to tolerate ambient heat, it is the outcome of the maintenance of thermal balance, but not only. Animal of low metabolic rate are thermotolerant, yet they are not a solution to climatic influences on production. Heat tolerance was shown to be correlated with low performance.   

The topic comprises a large variety of animals, differing in size, living in a wide range of habitats, located in widely differing climates, having a wide range of contributions to human societies that differ in their socioeconomic characteristics.

The review was expected to give attention to the impact of religions on the species of animals used in animal farming in the African continent and the Indian subcontinent. The resulting limitations are not examined.

The review points to the potential contribution of small mammals to animal farming in regions of low socioeconomic level, but does not elaborate on it, in spite of its importance.

The review stresses the importance of selection and proposes it for creation of heat tolerant animals (e.g. lines 297-9). But breeding programs or gene introgression and not selection are the adequate tools for producing stable genetic changes.

The review mentions a large number of traits for which genetic information was made available in a large range of species and breeds, in a variety of climatic and nutritional environments. Pooling them with no detailed examination does not provide much beyond mere information.  This would sassist in creating thermotolerance combined with significantly higher production.    

The traits for which genetic information was presented were not grouped in physiological sequences associations by function or by animal breeds.  The Bos taurus and Bos indicus do not consist of uniform groups of breeds. There are heat tolerant Bos taurus breeds.

Specific comments

Line

19        definition of heat stress is: the factors present in the environment that create a heat load. Change to : heat strain can be mninimised

30-1     it is stated :  “effective strategy may be  sustainable intensification of ruminant production.” This however increases metabolic heat production and heat strain. Correct or delete

63        “adapting to poor quality feed” ? this a new, untested  concept in nutrition

68        “animal of the future will need to be able to be more efficient by adapting to poor quality feed and changing climate without compromising performance”.  Evidence is needed to show presence of capacity adapt to poor quality feed.  Modify or delete

86-9     delete sentence after “intake”,  as it is not accurate.

90        delete “dramatic”, milk yield loss is moderate

94        delete after “temperature” to end of sentence. It is not logically related.

100-36 delete, it is not related to subject

139-41 references to content or delete

149-50 could you expand on this matter?  It is most important in particular for animal farming in the African continent.

Table 1. The table consists of sections containing detailed data of single experiments of different species. This presentation mode does not serve the purpose of a review.  Replace it by a textual evaluation of heat stress mitigation methods, or reduce the section to a minimum, as it does not add much to the theme of the review.

192-202 Pooling Zebu breeds and Bos taurus breeds each into one entitiy is wrong and is misleading, and as a result this part of the review is unacceptatble.  So are other parts of the review in which this generalization is present.  Also, part of the African breeds belong to the Bos taurus.

213-21 delete, the content is a number of general statements, that dos not belong to the context of the review

230-50 delete sections, as it consists of sentences missing a connecting logical link

Table 2. The table consists of an enumeration of studies’ data and is not a review and interpretation of their results. Replace the table by a suitable text.
